# Proximate determinants of the frequency of mosquito sounds: separating species-specific effects from environmentally driven variations - Implications for AI species recognition

Julie Augustin[1,2]*, Sándor Zsebők[2,3], Dorottya Kovács[1,2], Zoltán Jánki[2,4], András Bánhalmi[2,4], Zoltán Soltész[1,2], Péter Seffer[2,4], Vilmos Bilicki[2,4], László Zsolt Garamszegi[1,2]

1 Evolutionary Ecology Group, Institute of Ecology and Botany HUN-REN Centre for Ecological Research, Vácrátót, Hungary, 2 National Laboratory for Health Security, HUN-REN Centre for Ecological Research, Budapest, Hungary, 3 Department of Systematic Zoology and Ecology, Eötvös Loránd University, Budapest, Hungary, 4 Department of Software Engineering, University of Szeged, Szeged, Hungary

* julie.augustin@ecolres.hu

## Abstract

In recent years, several technologies have been developed for the monitoring and control of insect vector species. Many of them aim to use mosquito wingbeat frequency in the form of sound or opto-acoustic measurements to identify mosquito species, often through the training of AI classification models. However, these models often struggle to be accurate in real-life conditions, as the training data rarely captures the variability range of different species across many individual and environmental conditions, or does not explicitly control for it. Here, we use lab recordings of mosquito sounds to evaluate the impact of several environmental and life history factors on the mean frequency of the first harmonic of mosquito sounds. We recorded 475 individuals of 15 species in several environmental conditions, varying in temperature and humidity, while we also characterized the effect of body size (wing length), sex and age on the frequency of wingbeat sound at the among-individual level. Only species that comprised at least 2 recorded individuals were included in the analysis (N = 10 species). Variances at the within-individual and within-species level varied consistently, as the repeatability of the trait was 0.411 and 0.466, respectively. However, when we controlled for morphological and environmental effects, the proportion of between-individual variance decreased, while the between-species component increased (repeatabilities: 0.267 and 0.630). This suggests that species-specific signals in the sound are more robust once factors introducing variances due to real life conditions are involved in the models. Sex and temperature both had a significant effect on mosquito sound: an increase in temperature led to an increase in wingbeat frequency. In addition, the random slope analysis showed that response

**Data availability statement:** All data files are available from the figshare database: https://doi.org/10.6084/m9.figshare.30230581.

**Funding:** The project was funded by the Hungary's National Research, Development and Innovation Office (K135841, RRF-2.3.1-21-2022-00006, ADVANCED 152427) (https://nkfih.gov.hu/aboutthe-office) (LZG). The funders had no role in study design, data collection and analysis, decision to publish, or preparation of the manuscript.

**Competing interests:** The authors have declared that no competing interests exist.

to temperature differ between species, with strong between-species differences, especially for males. Therefore, advancing AI species recognition requires that biotic and environmental variables be either explicitly integrated into classification models or sufficiently represented in training data to reflect real-life variability.

## Introduction

Mosquitoes are vectors of several human and animal pathogens, including the malaria parasite and the dengue, chikungunya and ZIKA viruses, that are responsible for hundreds of thousands of deaths, and millions of cases every year [1]. The most effective way to cope with the threat of emerging or re-emerging vector borne diseases is the prevention by rigorous surveillance system, which can help early detection of risk and the initiation of mitigation efforts (*e.g.,* mosquito control).

There are hundreds of species of mosquitoes, but only a few of them can actually transmit pathogens to humans; for example, malaria is typically transmitted by mosquitoes of the Anopheles genus, most predominantly *An. gambiae* and *An. funestus* [2]. Similarly, dengue is mainly transmitted by *Aedes aegypti* and *Ae. albopictus* [3]. If it can be interesting from an ecological point of view to monitor all species of mosquitoes, from a public health perspective, only few species are relevant. One of the most important aspects of vector monitoring will therefore be to detect and identify these select vector species.

In recent years, numerous technologies have been developed to monitor and control vector and vector-borne diseases [4], including automatic remote monitoring [5], drones [6] and mobile-phone based citizen sciences [7–9]. Many of these tools rely on deep-learning technologies [5,10,11], specifically for the detection and classification of species. Models are often trained on visual data [12,13], or on acoustic or opto-acoustic data [14,15]. The visual data consists of pictures of the whole individual or some of its parts, typically wings. The acoustic data consists of recordings of the flying sound with a microphone. Finally, the opto-acoustic data indirectly measures the flying sound; the insect passes through a light beam and occults a sensor with its beating wings, allowing the measurement of its wingbeat frequency, directly related to sound.

Acoustic data in particular, used as passive acoustic monitoring, could allow the surveillance of vector populations in real-time [14], and assist timely public health decisions. Several classification models have been trained for mosquito sounds specifically, either based on the acoustic recording of the mosquito flying sound [16–20], or their opto-acoustic properties [21–26]. These models can reach high accuracies (up to 97%); however, these high accuracies are generally obtained with recordings realized in highly controlled lab conditions [25], and containing few species (typically 3–5, sometimes classified at the genus level instead of the species level) [27]. Accuracies are typically lower when more species are included (between 35% to 78% for 23 species) [16,17]. One practical caveat is the lack of publicly available datasets that includes many species for training the models (although more datasets have

become available in recent years) [28]. In addition, the implementation in the field remains complex due to the faint sound of mosquitoes (especially compared to anthropogenic noises). All these factors contribute to a reduced robustness of the classification models in real conditions; consequently, they remain largely underutilized in the field [28].

In addition, the sounds of wild mosquito populations are likely much more variable than those represented in the training data, which further reduces the applicability of AI-based species recognition in real conditions. Many environmental and biotic factors can influence mosquito sounds (Figs 1 and 2), and such information and variability are generally lacking in most training datasets. The origin of the mosquitoes and the recording conditions can largely affect the resulting mosquito sound. For example, if mosquitoes originated from an artificial colony and were recorded in the lab, large differences with wild populations could result from differences in size, age, feeding status, cage density, temperature, relative humidity and time of recording. Similarly, if different species are recorded in the field in systematically different environments, this will also create a bias, resulting in pseudo-species signals in the training data. All of these would affect species recognition models, as they would only learn to recognize a given species in a given environment, and wouldn't be able to generalize the species-specific signals. Notably, some studies successfully improved classification accuracy by including additional data such as the location [16] and timing of the recording [21,29,30]. However, to explicitly account for these factors, we need to know how they impact the different species that we aim to classify. There are several studies about the impact of biotic factors on mosquitoes' wingbeat frequencies, but unfortunately most of them focus on select model species (Fig 1). Furthermore, this type of data remains especially scarce for environmental factors (six studies on two model species) (Fig 2).

Figs 1 and 2 illustrate a strong bias towards few model species; primarily *Aedes aegypti*, and to a lesser extent *Aedes albopictus, Culex quinquefasciatus* and *Anopheles gambiae*. The focus on these species can be explained by i) their relevance as disease vectors (*Aedes aegypti* is the most important vector of yellow fever, dengue, chikungunya and ZIKA viruses [31–33] and ii) their relative ease to breed in the lab [34]. The impact of environmental factors on wingbeat frequency has been predominantly investigated in *Aedes aegypti*, with studies examining temperature, humidity, wind, atmospheric pressure, and light. Overall, wingbeat frequency increases with temperature, and at high temperatures, wingbeat frequency also increases with humidity (S2 Table). *Culex quinquefasciatus* has been studied to a lesser extent, only with time of day and population density. Because different environmental variables have been assessed in different species, species-specific responses cannot be directly compared (Fig 2). In contrast, the effects of biotic factors on wingbeat frequency have been examined across a wider range of species and studies (Fig 1). Some patterns are consistent: males exhibit higher wingbeat frequencies than females, mating status does not appear to alter wingbeat frequency, and mosquitoes display changes in wingbeat frequencies in response to specific incident sounds. By comparison, factors including age, size, oviposition, swarming, and feeding have been studied in multiple species, yet findings remain inconsistent across studies. For age, the general trend is that wingbeat frequency increases during the first few days of life before reaching a plateau, suggesting non-linear associations (S1 Table), although some studies have reported a further increase in the weeks following emergence from the pupal stage [35,36]. Wing length also influences wingbeat frequency, but trends vary both between and within species (S1 Table). Overall, these findings indicate that most of our knowledge on the effects of environmental and biotic factors on wingbeat frequency is derived from a few model species—primarily *Aedes aegypti*, and to a lesser extent *Ae. albopictus*, *Culex quinquefasciatus* and *Anopheles gambiae*—and even in these species, observed trends are not always consistent.

Mosquito sound is generally described as the wingbeat frequency [37], flight tone frequency [38] or fundamental wingbeat frequency [39], which fundamentally capture the same phenomenon. Because mosquitoes emit sounds when flapping their wings during flights [40], it is generally considered that the fundamental frequency (first harmonic) of their sound represents the direct frequency at which they beat their wings [41–43]. Both calculations on the theoretical amplitude of the sound based on the animal size and wingbeat frequency [44] and biologically relevant responses to these frequencies and amplitudes [45,46] tend to confirm this assumption. Therefore, to remain consistent with the literature of mosquito

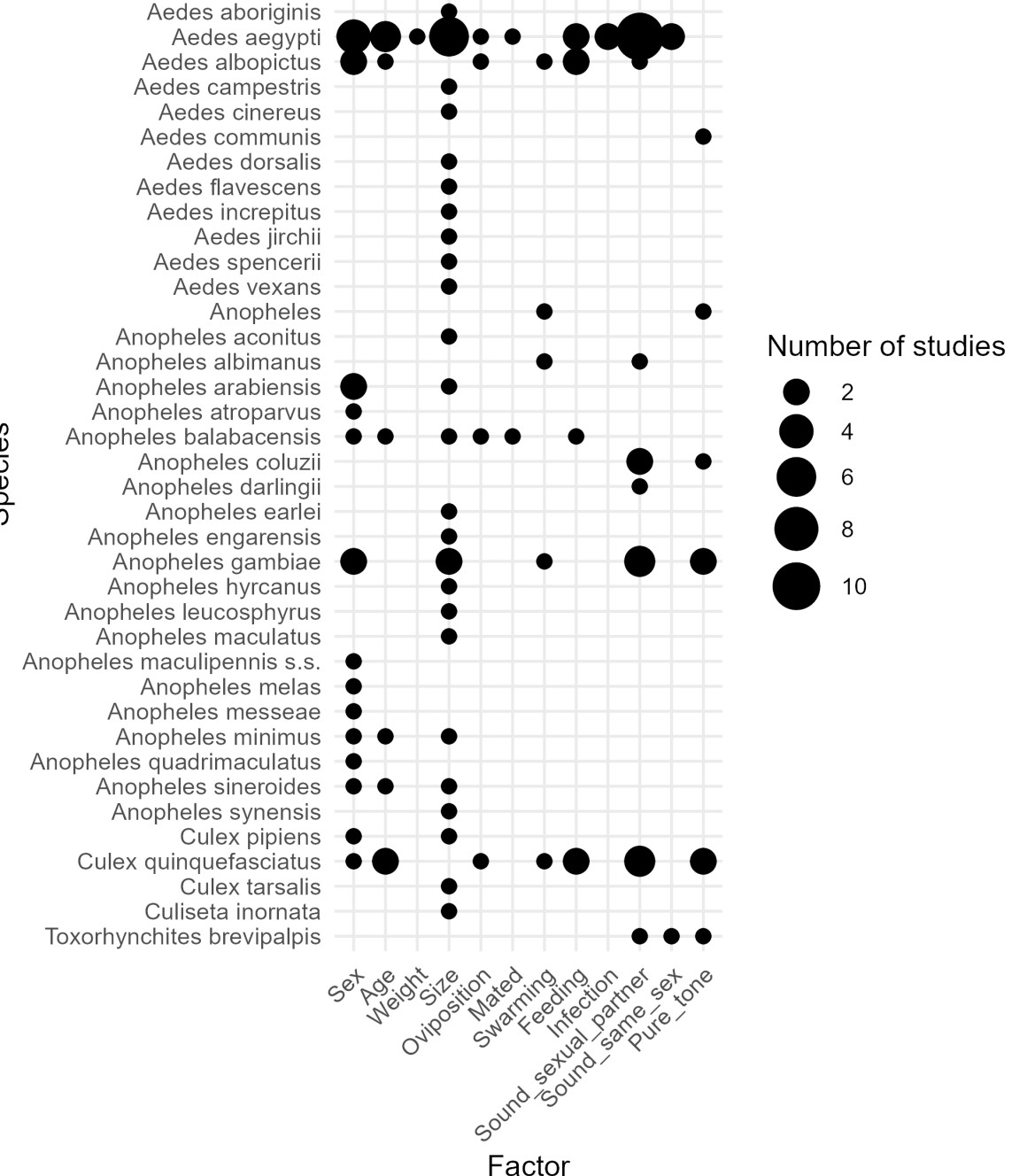

**Fig 1. Impact of biological factors on mosquito wingbeat frequency (WBF) (frequency of the first harmonic).** The data used for the creation of Fig 1 is available as supplementary material (S1 Table). While more than 30 species have been evaluated, most of them have only been investigated in one or two studies, with the exception of *Aedes aegypti*, *Aedes albopictus*, *Culex quinquefasciatus* and *Anopheles gambiae*. In addition, only the impact sex and size have been investigated in a high number of species, while other factors remain little investigated.

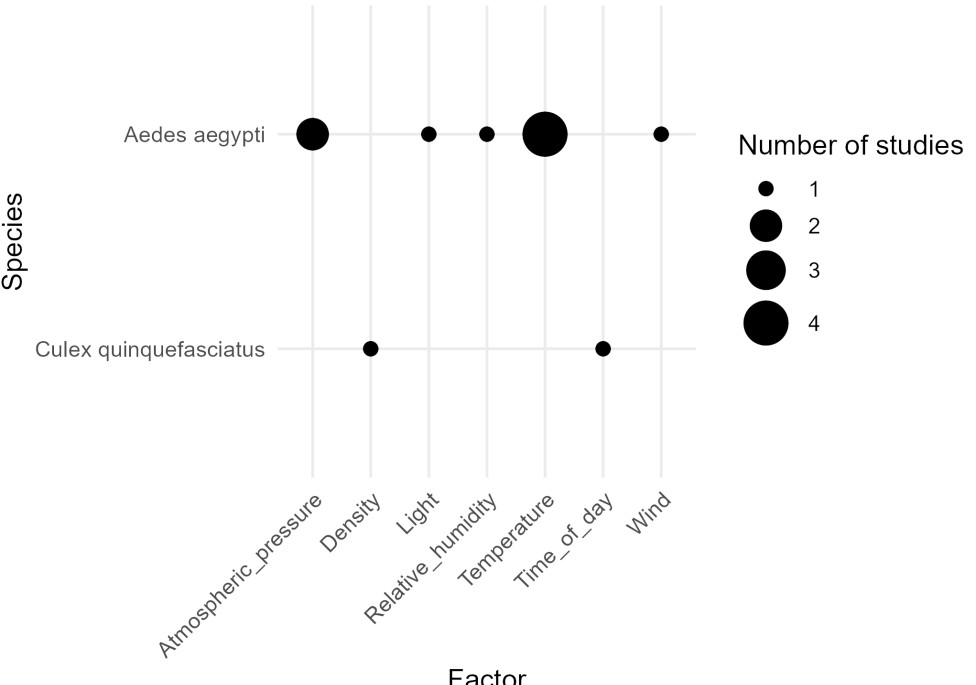

**Fig 2. Impact of environmental factors on mosquito wingbeat frequency (WBF) (frequency of the first harmonic).** The data used for the creation of Fig 2 is available as supplementary material (S2 Table). Only two species have been examined, in a total of six papers.

sounds, we will also adopt this terminology in this paper, referring to the frequency of the first harmonic as the wingbeat frequency.

Here, we used laboratory recorded mosquito sounds for 475 individuals from 15 species to demonstrate whether the determinants of the within- and between species variance can interfere with the species-specific signal in an acoustic trait, an effect that can have consequences for the development of AI technologies for species identification. To evaluate this, we calculated the repeatability of wingbeat frequency at both the individual and species levels, defined respectively as the proportion of between-individual or between-species variance relative to the total variance. For example, if repeatability is high (close to 1) at the species level, the trait is consistent among the species, and it will result in more robust AI-classification systems, because the species-specific signal is strong. In contrast, if repeatability is low (close to 0) at the species level, the signal is much more variable among a given species, and the classification systems will be far less robust or not able to perform at all. In addition, we tested the impact of the natural variation of environmental and biotic factors occurring in the recordings on sound variance. Due to its crucial role on insect physiology [47], and its documented impact on mosquito sound (Fig 2) we first investigated the impact of temperature. Second, because of its potential role on mosquito flight in relation to temperature (Fig 2) and its importance for mosquito thermal biology [48], we investigated the relative humidity (later referred as "humidity"). Third, due to the strong circadian variations in mosquito activity [49], we evaluated the impact of the time of recording. We also assessed the effect of three biotic factors, due to their documented impact on mosquito sound: sex, age and size (proxied by wing length) (Fig 1). Due to the impact of the recording conditions on the mosquito sound, we predicted that repeatabilities will be higher at the within-species level, and lower at the within-individual level when including the environmental and biological variability into the model. For the impact of the different variables, we predicted that all (temperature, humidity, time of day, sex, age and wing length) will have a significant effect on the mosquito wingbeat frequency, due to their documented impact on mosquito sounds. Given that different

studies revealed different relationships between wingbeat frequency and the investigated predictors, we also examined if the within-species associations can vary among species.

## Materials and methods

### Mosquito collection

Mosquitoes were collected from March to November 2024 (N = 475), from several locations in Hungary, using ovitraps (eggs, N = 47) and dip nets (larvae, N = 402). No adults were collected from the field, only egg and larvae stages. Because we only sampled invertebrates, did not sample from private lands or protected sites, nor sampled any protected species, no permit was required. Seventeen locations were chosen based on previous species occurrence data, to obtain as many species as possible (S3 Table). Eggs and larvae were bred in the mosquito facility at the Institute of Aquatic Ecology of the HUN-REN Centre for Ecological Research, in Budapest. In addition to the field collection, we used *Aedes albopictus* mosquitoes from the permanent colony (N = 26), which was created using eggs from the National Centre for Public Health and Pharmacy (NCPHP) colony in Budapest, Hungary, and from the FAO/IAEA in Vienna, Austria. All colonies were established and maintained following FAO/IEAE guidelines: individuals were kept in 30x30x30 cm Bugdorm cages, at 26ºC ± 1ºC and 70 ± 5% humidity, with a 12h:12h light:dark photoperiod including dusk (1hour) and dawn (1hour) [34,50,51]. 10% sugar water was available *ad libitum*, but mosquitoes were not blood-fed. *Aedes albopictus* individuals reared from laboratory colonies did not differ in wingbeat frequency compared to the individuals collected from the field, and were therefore included in the analysis ($\chi^2(1) = 0.03$, p = 0.86). The mosquito breeding facility followed the arthropod containment guidelines (Level 1) for uninfected arthropods already present in the local geographical region [51,52]. This study followed all international, national and institutional ethical regulations and guidelines; because this study only involved invertebrates, no ethical permit was required. This was approved by the HUN-REN Center for Ecological Research ethical committee.

Because all individuals from the same origin did not emerge synchronously in the same cage, it was not possible to know the exact age for each recorded individual. For each cage and each day of recording, we assessed the minimum and maximum age of the individuals based on the dates of first and last known adult emergence, from which we calculated the median ages (in days). The median is robust to skewness of the emergence distribution, and does not rely on assumptions about the distribution shape. This method was applied identically across all cages and recording days, which ensured consistent and comparable age estimates between species and origins. We later used the range between minimum age and maximum age to account for the estimation uncertainty of age in the statistical model (see below).

### Acoustic recording

Mosquitoes were recorded one at a time. Individual adults were gently collected from their breeding cage using a custom-made mouth aspirator. Because tethering can affect the mosquitoes' flying sound [53], individuals were recorded in free flight. For each recording, one mosquito was transferred into a cage made out of mosquito net (10x10x10cm), which was placed inside a soundproof box (100x50x50cm). The cage size represented the best trade-off between the space allowed for mosquitoes to fly without constraint, and the distance between the mosquito flying in the cage and the microphones (which needed to be small for the sound to be detected). The cage was big enough for the small and medium species to fly freely, but it may have affected the flight of the biggest species. The inside of the soundproof box was lit using a LED light. The mosquito was given 15 minutes to recover from the transfer and acclimate to the recording conditions. Individuals that showed signs of injuries after the transfer (damaged wings, not moving when stimulated, or jumping instead of flying) were removed from the experiment (N = 12). Because low temperatures can prevent flying [54], a minimum temperature of at least 20°C was maintained in the soundproof box with a terrarium heating cable (maximum temperature recorded = 31.5°C). After acclimation, the mosquito was recorded for 10 min using one of two

recording set-ups. The first set-up consisted of 4 microphones (CMP5247TFK) (one on each side of the cage) connected to pre-amplifiers (based on [55] and a Zoom H4n Handheld Digital Recorder (sample rate: 48kHz)). The second set-up consisted of 4 Audiomoth devices [56], placed on the sides of the cage (sample rate: 48kHz). The recording equipment had no significant effect on the fundamental frequency of the mosquito sound ($\chi^2(1) = 0.19$, $p = 0.67$), thus we combined sound recordings regardless of the equipment used. Every 30s, if the mosquito was not already flying, it was stimulated to fly by tarsal contact: the experimenter reached inside the soundproof box through an opening on the side, and gently brushed the net underneath the mosquito tarsi with a small spatula [57]. Despite the opening, the soundproof box reduced external noise and enables higher-quality recordings of the mosquito sounds. There was a strong variability in willingness to fly between individuals, with some mosquitoes flying non-stop throughout the 10 min period, and some flying only for a second when stimulated. We included all mosquitoes that were observed flying in the analysis, regardless of their flight duration during the trial.

## Ecological predictors: temperature, humidity and time of day

As noted earlier, we used a terrarium heater to keep the temperature above 20 °C. Beyond this, temperature and humidity were not controlled in the laboratory, leading to considerable seasonal variation (Table 1). Temperature and humidity were measured inside the box during the recordings using a data sensor and logger (Voltcraft DL-210TH). The time of recording (time of day) was also noted for each recording. Unfortunately, we were limited to one breeding room with one photoperiod, so all mosquitoes had to be recorded during the day, irrespective of the species circadian rhythm. The photophase in the breeding room started at 6:00 (dawn) with the light reaching full intensity at 7:00. Recordings thus occurred all throughout the photophase, but never during the scotophase (see Table 1 for the range of recording time).

## Wing length measurements and age

After recording, mosquitoes were transferred individually into Eppendorf vials and euthanized in the freezer (−14.5°C). Specimens were then identified to the species by a taxonomist expert, and their sex was determined using morphological characteristics. The right wing of each individual was detached and mounted on a microscope slide for measurements. Wings were later photographed using the Toupview software (https://www.touptekphotonics.com) and a C2CMOS12000KPA camera mounted on a microscope. Several pictures were taken on each wing, and were stitched together using the Microsoft Image Composite Editor. Wing length was then measured using the ImageJ software (https://imagej.net/ij/), from the axillary incision to the apex of the wing, excluding the fringe setae [58]. Because the wing length was intrinsically dependent upon the species, we standardized it using (individual wing length – average wing length for the species)/ standard deviation of wing length for the species. We expected that the raw wing lengths would disproportionately drive the between-species wingbeat frequency. Therefore, we used the standardized version to properly evaluate the impact of the other predictors, and to better estimate the impact of wing length at the within-species level.

**Table 1. Range for the predictors included in the model (except sex which was binary).**

| Predictor | Mean ± SD | Min | Max |
|---|---|---|---|
| Temperature (°C) | 24.5 ± 2.0 | 20.5 | 31.5 |
| Humidity (%) | 54 ± 11 | 30 | 71 |
| Time of day (hour:minutes) | – | 07:37 | 17:07 |
| Wing length (mm) | 3.4 ± 0.8 | 1.6 | 5.8 |
| Age (number of days since adult emergence) | 13.8 ± 12.3 | 0.5 | 49 |

## Automatic detection of mosquito sounds

Mosquito sounds were analyzed blindly to the species identity and to the recording conditions. Individual mosquito sounds were detected in each recording using a custom-made mosquito detection module, implemented in Python (https://github.com/MosquiTUNE/Mosquito_sound_detection_methods/tree/main). For each recording, the model identified 1-second audio segments as containing or not containing mosquito sounds, and for those that contained mosquito sound, the model provided the timestamps from the original wav file.

## Acoustic processing and measurements

Recordings were randomized in their order prior to acoustic processing to minimize bias associated with processing sequence. Automatically detected mosquito sounds were checked manually using Raven Pro 1.6 (time window: 2s; FFT-window: 2048). Because each recording contained 4 audio channels, if the same mosquito sound was present in more than one channel, the best quality one was selected. Better quality was defined as a longer sound, no or little fragmentation, and a high amplitude of the sound compared to the background noise. Only sounds that were at least 0.2s were selected, and the time frame between two sounds for them to be considered distinct was 1s. We aimed to select at least 10 sounds per recording, but only those that reached the minimum quality standard were selected, resulting in 1–25 sounds per recording (8.7±3.3, mean±SD). If only one high-quality sound was available for the whole recording, only this sound was kept; low-quality sounds were not included in the analysis. The selected sounds were evenly temporally distributed within the recordings. The acoustic measurements were conducted in R using the "soundgen" package [59] via a custom-written script (https://doi.org/10.6084/m9.figshare.30230581). The frequency track of the first harmonic was identified using the "analyze" function within the manually selected regions. We applied the following settings: a 2048-point Hanning window for the FFT, with a 25% window overlap. For each window step, the pitch of the harmonic was determined using the "analyze" function with the "spec" and "zc" algorithm settings. We then computed mean frequency of the first harmonic for each sound (extracted output parameter=frequency_mean). Altogether we were able to obtain sound recording and perform acoustic analyses for 15 mosquito species, and most species were represented with more than one individual (Table 2).

Table 2. **Number of individuals, recordings and individual sounds obtained per mosquito species.**

| Species | N individuals | N recordings | N sounds |
|---|---|---|---|
| *Culex pipiens* | 186 | 193 | 1685 |
| *Aedes albopictus* | 105 | 108 | 905 |
| *Aedes koreicus* | 60 | 62 | 616 |
| *Ochlerotatus geniculatus* | 35 | 37 | 268 |
| *Aedes japonicus* | 29 | 29 | 274 |
| *Aedes vexans* | 27 | 27 | 234 |
| *Ochlerotatus annulipes* | 15 | 15 | 114 |
| *Culiseta longiareolata* | 7 | 7 | 66 |
| *Culex hortensis* | 4 | 4 | 42 |
| *Ochlerotatus rusticus* | 2 | 2 | 14 |
| *Aedes sticticus* | 1 | 1 | 10 |
| *Anopheles claviger* | 1 | 1 | 11 |
| *Culex modestus* | 1 | 1 | 3 |
| *Culiseta annulata* | 1 | 1 | 10 |
| *Culiseta morsitans* | 1 | 1 | 9 |

The numbers represent all recorded individuals, but only the species that contained at least 2 individuals were included in the analysis.

## Statistical analysis

To determine the amount of variance that can be attributed to different hierarchical levels, we constructed a linear mixed effects model with the appropriate random effect structure and using Gaussian error distribution. The response variable was the mean frequency of the first harmonic (log10-transformed in order to fulfill criteria for the distribution of residuals), and the descriptive part of the model only included the intercept (no fixed effects were entered). We defined random effects based on the individual ID (since usually more than one sound was analyzed for the same individual), the site of origin and species ID (the origin of the specimen was defined as the place of collection if the specimen originated from the field, and the colony of origin if it originated from an existing colony). From the fitted model, we extracted variance components corresponding to these random effects, and we calculated the repeatability of the focal trait (wingbeat frequency) at different levels (individuals, origin and species) by dividing the respective variance components by the total variance (species variance + origin variance + individual variance + residual variance). The repeatability metric measures how consistent the focal trait is at the studied level (for example how consistent the wingbeat frequency is within individuals compared to between individuals). The 95% confidence intervals of the repeatabilities were determined based on parametric bootstrap methods [60] (N = 1000 simulations).

In a second model, we entered wing length, age, sex and environmental variables to examine how they affected the above repeatability estimates (i.e., to investigate how the control for these effects enhanced the individual- or species-specific signal in the data). Accordingly, we defined the following fixed effects: sex, temperature, humidity, wing length (standardized per species), median age (number of days elapsed since hatching, calculated as the median age for a given cage), and time of day of the recording (as numeric values). Because the date of the recording was highly correlated to both temperature ($r(449)$ = −0.38, $p < 0.001$) and humidity ($r(442)$ = −0.61, $p < 0.001$), we removed it from the fixed effects. The random part of the model was the same as defined above (species ID, the origin, and the individual) as random effects, and the repeatabilities from these models were also calculated as defined above. The significance of fixed predictors was determined based on Wald chi-square tests. Fixed predictors were centered and standardized. When evaluating the effect of age, we re-fit the model that included weights as 1/sqrt(age variance) to account for uncertainty that is associated with the estimation of age.

Third, we also examined if the significant fixed predictors were acting similarly in different species or if their corresponding slopes varied among species. This was done by fitting random slope models that also estimated the variance of regression slopes among species. To check if the random slope model offered better fit to the data than the model including only random intercept, we compared these models with likelihood ratio test. We did not examine the impact of significant fixed predictors within individuals, because most individuals were only recorded once, so the fixed predictors values were the same for all sounds of a given individual.

Table 2 represents the individuals for which we obtained sounds. However, only species that contained at least two recorded individuals were included into the analysis; so over 15 species collected, only 10 were included into the mixed models. In addition, because in some instances some variables were missing (*e.g.,*: age, wing length), the sample size varied across models.

All analyses were carried out in the R statistical environment [61], using R studio (2023.12.1 Build 402). The mixed models were fit using the lmer() function from the lme4 package [62]. Wald chi-square tests to estimate the significance of fixed predictors were calculated using the Anova() function from the car package [63].

## Results

We were able to obtain good quality recordings and identify the species for 475 individuals, from all 15 collected species (Table 2).

For more than half of the species, this is the first documented measurement of their wingbeat frequency (Table 3) (*Aedes koreicus, Aedes sticticus, Anopheles claviger, Culiseta annulata, Culex modestus, Ochlerotatus geniculatus, Ochlerotatus rusticus, Ochlerotatus modestus*) [64,65].

**Table 3. Wingbeat frequencies of the collected species.**

| Species | Wingbeat frequency (Hz) | | |
| --- | --- | --- | --- |
| | Females | Males | Combined |
| *Aedes albopictus* | 517±62 (515; 320–672) | 687±113 (694; 410–901) | 562±109 (540; 320–901) |
| *Aedes japonicus* | 338±32 (338; 271–428) | 503±56 (504; 414–612) | 370±75 (345; 271–612) |
| *Aedes koreicus* | 353±35 (354; 245–484) | 521±69 (516; 359–685) | 386±81 (369; 245–685) |
| *Aedes sticticus* | 266±9 (268; 254–280) | – | – |
| *Aedes vexans* | 345±46 (350; 181–430) | 464±55 (487; 379–532) | 363±64 (362; 181–532) |
| *Anopheles claviger* | 377±8 (374; 362–390) | – | – |
| *Culex hortensis* | 419±8 (420; 404–436) | 479±81 (484; 385–573) | 448±63 (422; 385–573) |
| *Culex modestus** | 309±14 (317; 293–318) | – | – |
| *Culex pipiens** | 324±36 (324; 192–414) | 578±77 (582; 293–771) | 424±136 (355; 192–771) |
| *Culiseta annulata** | – | 388±9 (390; 372–405) | – |
| *Culiseta longiareolata** | 243±33 (249; 179–306) | 297±41 (306; 251–365) | 255±41 (253; 179–365) |
| *Culiseta morsitans* | 168±5 (167; 160–174) | – | – |
| *Ochlerotatus annulipes* | 245±55 (233; 198–470) | 305±63 (317; 205–409) | 262±63 (237; 198–470) |
| *Ochlerotatus geniculatus* | 358±36 (356; 259–426) | 507±39 (508; 447–610) | 399±76 (386; 259–610) |
| *Ochlerotatus rusticus* | 305±12 (303; 288–322) | 335±6 (333; 328–345) | 320±18 (325; 288–345) |

Values are presented as mean±sd (median; range). All mosquitoes were recorded during the day. Species that are known to generally be active at night are indicated with a "*", and their wingbeat frequencies might differ slightly if they had been measured during their standard activity period. For some species the activity period is not well-known and thus could not be assessed here (*Aedes sticticus*, *Culex hortensis*, *Culiseta morsitans*). Standard deviation values are somewhat high (>10% of the mean), but not unexpected, as wingbeat frequencies fall within a broad range, even within the same species and sex [64]. In addition, in some rare occasions, variations of up to 100 Hz were observed for a single individual within a single recording, displaying the large intra-individual range of the trait (we always checked the original data to verify that these extremes are not due to some data errors or some non-biological constraints).

## Repeatability analysis

If wingbeat frequency bears with any species- or individual-specific signal, it should have non-zero repeatability. This is what we observe here, indicating that wingbeat frequency has some basis in both species and individuals. The model that included no fixed predictors (only intercept) estimated the highest repeatability at the within-species level, but within-individual variation was also relatively consistent (Table 4). This means that, when we did not include the environmental and biological effects, the wingbeat frequency was very consistent among species (the wingbeat frequency varied little between individuals of the same species compared to the total variation). The trait was almost as consistent among individuals (the within-individual wingbeat frequency varied little compared to the total variation). Interestingly, when adding environmental predictors and sex, wing length and age to the model, the repeatability increased at the species level, and individual repeatability decreased. This means that the species signal became clearer, suggesting that some of the previously observed variance in wingbeat frequency was caused by these external factors, and not by signal variation within the species.

## Predictors of wing beat frequency

We tested the impact of sex, temperature, humidity, wing length (standardized per species), age, and time of recording. Sex and temperature both had a significant effect on the wing beat frequency (Table 4). Females had lower wingbeat frequencies than males, while wingbeat frequency increased with rising temperature. The effects of other predictors entered in the model were not significant.

The effect of a predictor can be manifested in the same way in different species (i.e., individuals of any species measured at higher temperature have systematically higher frequency for their sounds due to some physical constraints) or

**Table 4. Results of the mixed model.**

| | Intercept only | | | | With fixed predictors | | | | | |
|---|---|---|---|---|---|---|---|---|---|---|
| **Random effects** | | | | | **Random effects** | | | | | |
| | **Fitted model** | | **Repeatability analysis** | | | **Fitted model** | | **Repeatability analysis** | | |
| **Groups** | **N** | **Variance** | **Repeatability** | **95% CI** | **Groups** | **N** | **Variance** | **N** | **Repeatability** | **95% CI** |
| individual | 464 | 0.0089 | 0.411 | [0.252; 0.654] | individual | 263 | 0.0029 | 284 | 0.267 | [0.153; 0.533] |
| origin | 18 | 0.0025 | 0.117 | [0.025; 0.248] | origin | 15 | 0.0019 | 17 | 0.091 | [0.010; 0.245] |
| species | 10 | 0.0100 | 0.466 | [0.167; 0.677] | species | 9 | 0.0059 | 10 | 0.630 | [0.283; 0.789] |
| Residual | | 0.0001 | | | Residual | | 0.0001 | | | |
| **Random effects** | | | | | **Random effects** | | | | | |
| | Estimate | Std. Error | t value | | | Estimate | Std. Error | t value | Chisq | Df | p-value |
| (Intercept) | 2.5406 | 0.0355 | 71.63 | | (Intercept) | 2.5285 | 0.0291 | 86.85 | | | |
| | | | | | sex | 0.1789 | 0.0119 | 14.99 | 224.542 | 1 | < 0.001 |
| | | | | | temperature | 0.0124 | 0.0048 | 2.58 | 6.668 | 1 | 0.01 |
| | | | | | humidity | −0.0093 | 0.0060 | −1.54 | 2.378 | 1 | 0.12 |
| | | | | | wing size | −0.0059 | 0.0055 | −1.07 | 1.138 | 1 | 0.29 |
| | | | | | age | 0.0088 | 0.0060 | 1.47 | 2.159 | 1 | 0.14 |
| | | | | | time of day | −0.0009 | 0.0035 | −0.26 | 0.069 | 1 | 0.79 |

different species may respond differently to elevating temperatures in terms of vocalization (S2 Table). To test for these scenarios, we fitted models (separately for males and females) that allowed the regression slopes to vary across species. This random slope model offered a better fit to the data (model for females: P < 0.001, model for males: P = 0.008, Fig 3) indicating that temperature will affect wingbeat frequency differently depending on the species. Mixed models are robust to variations in sample size, so the low number of individuals in some species should not generate bias in the analysis, although there could be some uncertainty around slope estimates. The slope of the regression lines was used to estimate the increase in wingbeat frequency (Hz) for every increase in 1°C [66]. (Table 5).

In females of all species, and in males of most species, wingbeat frequency increased with increasing temperature. However, males of three species: *Aedes japonicus*, *Aedes vexans* and *Culiseta longiareolata*, displayed a negative slope (although most of these groups had low N). Even for groups that had large number of individuals, the slope values varied between species and sexes. Most species displayed different slopes between males and females, although for some species the difference was very small; for example with *Ae. albopictus* and *Culex pipiens*.

## Discussion

We assessed the impact of environmental and biological variables on mosquito wingbeat frequencies, at the species and the individual levels, including a good number of non-model species (N = 10 analyzed). We found that i) within-species and within-individual repeatability was 0.466 and 0.411 respectively ii) repeatability changed when controlling for confounding effects, iii) sex and temperature had significant effects, iv) species responded differently to rising temperatures.

### Factors affecting mosquito sound

**Sex.** Males had higher wingbeat frequencies than females, which is consistent with what previous studies measured. The range of frequencies observed for most species is generally consistent with the literature [64]. In many mosquito species, females are larger than males [67], which likely affects the resulting sound, as bigger animals usually have lower

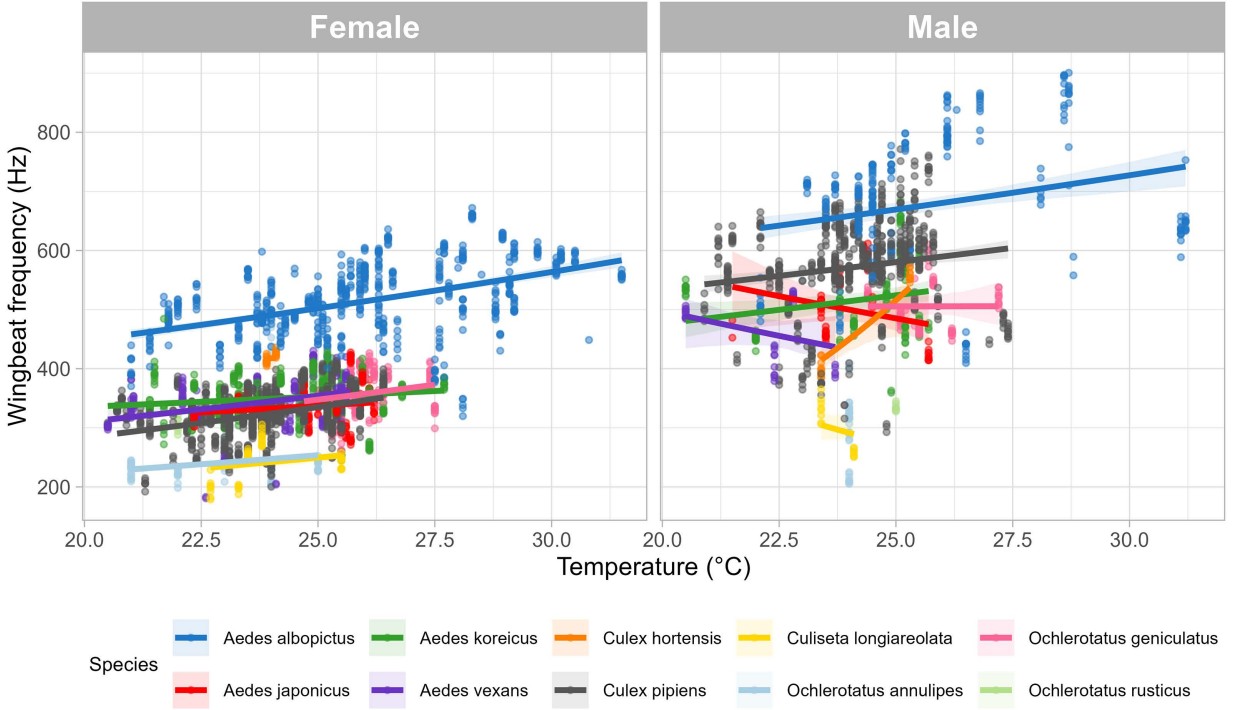

**Fig 3. Impact of temperature on wing beat frequency of different mosquito species shown separately for females (F) and males (M).** Slopes were computed for each sex and species using a linear regression (lines), that are shown with the 95% confidence interval (grey shaded areas). Different colors are for different species, dots are separate sounds.

**Table 5. Increase in wingbeat frequency for each supplementary °C for all analyzed species and sexes.**

| Species | Females | | Males | |
|---|---|---|---|---|
| | N | slope (Hz.°C⁻¹) | N | slope (Hz.°C⁻¹) |
| *Aedes albopictus* | 75 | 12 | 28 | 11 |
| *Aedes japonicus* | 22 | 5 | 7 | −15 |
| *Aedes koreicus* | 46 | 4 | 14 | 10 |
| *Aedes vexans* | 23 | 9 | 4 | −16 |
| *Culex hortensis* | 2 | 10 | 2 | 65 |
| *Culex pipiens* | 112 | 11 | 74 | 9 |
| *Culiseta longiareolata* | 5 | 7 | 2 | −22 |
| *Ochlerotatus annulipes* | 9 | 6 | 6 | NA |
| *Ochlerotatus geniculatus* | 25 | 10 | 9 | 0 |
| *Ochlerotatus rusticus* | 1 | NA | 1 | NA |

Slopes were extracted from the random slope model

sounds [68–70]. Moreover, the mosquito mating system heavily relies on sound, and the acoustic differences between males and females is key to the mate finding and recognition behaviours [46].

**Temperature.** Because insects are ectotherms, all of their physiology is affected by ambient temperature [71], including wingbeat frequency, as evidenced in both this paper and the literature (Fig 2). Our analysis revealed that the wingbeat frequency response to temperature is species-specific in mosquitoes.

While low sample sizes could explain some differences, in species with large datasets, the variation likely reflects biological effects (e.g., *Culex pipiens*, *Aedes albopictus*, *Aedes koreicus*). Temperature affects insect physiology through enzyme conformation and reaction rates [72]. A given trait typically increases with temperature, reaches an optimum, and then declines as temperature continues to rise, due to the breakdown of chemical reactions; following the thermal performance curve [73]. For the flying behaviour specifically, there is a direct relationship between temperature and flight muscle contraction rate [74–76]. Different temperature-dependent relations likely arise from adaptation to environments with varying temperature ranges and fluctuations. In tropical regions, temperatures tend to be relatively stable, and tropical or subtropical species such as *Aedes albopictus* typically exhibit a narrower thermal tolerance [77]. In contrast, temperate species— like *Culex pipiens*, *Aedes koreicus*, and *Ae. japonicus*—generally have broader thermal ranges. As a result, their response to a given temperature increase is expected to be less steep than that of tropical species. Consistent with this expectation, we found that the slopes for females *Cx. pipiens*, *Ae. japonicus*, and *Ae. koreicus* (11, 5, and 4 Hz °C$^{-1}$, respectively) were lower than for *Ae. albopictus* (12 Hz °C$^{-1}$). Species-specific effects of temperature could also originate from the host preference, due to their blood temperature [78]. Here, *Cx pipiens*, which feeds preferentially on birds, had a higher slope compared to *Ae. japonicus* and *Ae. koreicus*, which feeds preferentially on mammals [79]. Whether it is due to their habitat or biology, sexes and species respond differently to temperature; and in order to include temperature dependency into classification models, these thermal parameters should be known for each individual species. The thermal biology and ecology of certain mosquito species—such as *Aedes aegypti* [80], *Aedes albopictus* [80–83], and key vectors in the genera *Anopheles* [84] and *Culex* [85] —have been investigated extensively. Yet, the influence of temperature on flight behaviour in general and wing-beat frequency specifically remains poorly understood for most species (Fig 2) [80].

In addition to the species-specific differences, our results suggest sex-specific differences in the response to temperature. However, due to the small N in many groups, further studies should confirm this finding. If they are confirmed, sex-specific responses to temperature could be explained by size differences (males are typically smaller than females of the same species), or by distinct thermoregulatory needs of males and females. Only females feed on blood, which— especially when taken from endotherms like mammals and birds, or from basking reptiles—is typically hotter than the mosquito itself. To cope with this, females have evolved specialized thermoregulatory systems that are activated after blood feeding [86–90]. A different response to temperature in females and males also raises the question of their acoustic communication. Males typically detect the females by hearing their flying sound, and males' antennae and acoustic neurons are minutely tuned to the product of females' wingbeat and their own [46,91]. If temperature, as suggested, differently impacts females and males, would the potential mates still be able to hear and recognize each other? Some evidence suggest that males' response and the tuning of their antennae follow the same increase rate as female's wingbeat frequency [92,93]. Lapshin and Vorontsov further suggest that males being tuned to the product of their own wingbeat with the female's makes the process more resilient to temperature changes, although this was stated under the assumption that temperature affects similarly females and males wingbeat frequency [94], which remains to be clearly established. This crucial topic warrants further investigations.

**Non-significant factors.** Some factors that had a significant impact on mosquito wingbeat frequency in the previous studies (Figs 1 and 2) were not significant here (humidity, time of recording, age and wing length). Differences in methods could explain some (*e.g.,* time of recording), as well as uncontrolled confounding effect (*e.g.,* wing area, that is directly correlated to wingbeat frequency in bumble bees [95]); and in general our predictors' range was smaller, and followed natural variation rather than being controlled (*e.g.,* humidity). In contrast, our age variation was large, and was only represented by the median value, which, although weighed, introduced uncertainty. In addition, most of the documented variation in age occurs in the first 2–3 days of life [35,39,93,96], and most of our mosquitoes were older.

Interestingly, we did not find an effect of the wing length once it has been standardized per species. Wing length is intrinsically linked to the species, and would drive by default a large part of the variance observed in the wingbeat frequency. Standardizing it allowed us to properly evaluate the impact of other predictors at the between-species level, but

also to better estimate the wing length effect at the individual level. Our results showed that wing length does not seem to impact wingbeat frequency within-species (the small individuals of a given species do not necessarily exhibit higher wingbeat frequencies compared to the large individuals of the same species). This would point toward a species-specific wingbeat production mechanism rather than just a size-based mechanism. De Nadai et al. [37] found that wingbeat frequency decreased when wing length increased in *Ae. aegypti*. In contrast, Wekesa et al. [97] found the opposite trend in *Anopheles gambiae* and *An. arabiensis*. Different within-species relationships between size and wingbeat frequency could explain our negative results with the standardized wing length. As many papers show (S1 Table), when wing length is not standardized within species, there is generally a significant negative relationship between wing length and wingbeat frequencies (species that have longer wings have lower wingbeat frequencies). We expect to obtain the same results in our data with the raw wing length value (S5 Fig).

Because we were limited to one breeding room with one photoperiod, we had to record all mosquitoes during the day, irrespective of the different species circadian rhythm. This is a limitation of our study, as Kim et al. [39] showed that wingbeat frequency differed between the day and the night in *Culex quinquefasciatus*. The wingbeat frequency of nocturnal species may therefore by slightly different to what would have been measured during their regular activity period. Most of the species studied here were diurnal, but this could still affect our results.

Finally, since most predictors were previously tested in single species studies, and often with *Ae. aegypti,* our negative results could also mean that there are no general rules for these predictors, and that their effects can vary largely from species to species. If species respond differently to a given predictor, the species-specific slopes with different magnitudes and sign blur each other's effects resulting in a zero overall slope. Our results therefore stress that trends obtained in largely studied model species are not necessarily generalizable to other species.

## Within-species and within-individual variation in acoustic traits

While intra- and inter-specific and individual variation in acoustic traits are frequently assessed in vertebrates, including birds [98] anurans [99], and mammals [100], it is rarely evaluated in insects, despite insect acoustics constituting a strong research field for many decades [28,101–103]. A few examples exist in a handful of insect groups [104–107], but this is far from widespread. Here we show that acoustic signals in mosquitoes vary consistently at the between species level, and that environmental factors affect this response. Compared to other behavioral traits, the repeatability estimates are rather high [108]. High repeatability in species-specific signals is advantageous for AI-based species recognition, as lower within-species variability can reduce the amount of training data required to achieve accurate classification.

## Sound-based monitoring and AI

In recent years, many sound-based classification models have been developed [28,109], including for mosquitoes [16,18,29]. Yet, despite the display of high accuracies, most remain underutilized in the field [28]. Variability among a given species, due to either natural variability or environmental conditions, could limit model accuracy in real conditions. Our data show that the models using species-specific signals in the sound are more robust once factors introducing variances due to real life conditions are held constant (although our recordings were made in a controlled lab environment, which still underrepresents variations observable in the wild). This demonstrates that we cannot ignore intra-specific and intra-individual variability for AI based acoustic classification. One solution for better integration of natural variance would be to adequately represent that environmental and biological variability in the training data. Unfortunately, such complete databases remain rare, especially for invertebrates [110], and building these extensive databases would require a lot of time and effort. Alternatively, classification systems could control for or include additional environmental information to improve classification accuracy. As shown in this paper, temperature clearly affects wingbeat frequency, and needs to be accounted for when building acoustic-based species classification models. For example, Saha et al [111] proposed

a general statistical tool for correcting wingbeats frequencies of flying insects according to temperature, tool that could be included into the classification models. Similarly, some models improved classification accuracies by incorporating metadata in their training, like location and time of recording [16,21,29,30]. However, this also requires a deep knowledge about individual species' response to the given predictor. In all cases, in order to improve classification models' accuracy in real life conditions, and the chance that we can use them for monitoring purposes, we need to better understand and account for natural variability in the target populations.

### Perspective on mate selection and individual quality

In addition to the relevance for AI-classification, this research also presents findings with behavioural ecological importance. In mosquitoes, sound is the main cue used to find and recognize mates [46]. Therefore, high repeatability in acoustic signals can have consequences both at the species and the individual level. At the species level, high signal repeatability may enhance discrimination between species, reducing the likelihood of unsuccessful interspecific matings, and potentially contributing to reproductive isolation and speciation. For example, males of *Aedes aegypti* responded more strongly to the sound of conspecific females than to the sound of *Aedes albopictus* females [112,113]. At the individual level, a high degree of signal repeatability may allow traits to serve as an indicator of individual quality, and consequently influence mate choice decisions. Finally, some degree of consistent among-individual difference might be needed for maintaining the ability of the species to respond to different environmental conditions. If a population consists of non-uniform individuals, there is a higher chance that one individual can cope with a new environmental condition [98]. However, since this acoustic signal is directly related to the mosquitoes' flight ability, the trait is likely under strong mechanistic selective pressure, and can only display limited variation both within individuals and across species.

### Supporting information

**S1 Table. Biological factors that affect mosquitoes wingbeat frequencies.** WBF = wingbeat frequency. The mention of harmonic convergence and rapid frequency modulation follows the terms used by the authors in the papers.
(XLSX)

**S2 Table. Environmental factors that affect mosquitoes wingbeat frequencies.** WBF = wingbeat frequency.
(XLSX)

**S3 Table. Collection points of the eggs and larvae used in the study.**
(XLSX)

**S4 Table. Results of the mixed model and repeatability analysis with same number of lines for each model (N=263).**
(XLSX)

**S5 Table. P-values of the random slope analysis for the non-significant predictors.**
(XLSX)

**S1 Fig. Impact of humidity on wingbeat frequency.** F = females, M = males.
(TIFF)

**S2 Fig. Impact of time of recording on wingbeat frequency.** F = females, M = males.
(TIFF)

**S3 Fig. Impact of age on wingbeat frequency.** F = females, M = males.
(TIFF)

**S4 Fig. Impact of standardized wing length on wingbeat frequency.** F = females, M = males.
(TIFF)

**S5 Fig. Impact of wing length on wingbeat frequency.** F = females, M = males.
(TIFF)

## Author contributions

**Conceptualization:** Julie Augustin, Sándor Zsebők, László Zsolt Garamszegi.

**Data curation:** Julie Augustin, Sándor Zsebők.

**Formal analysis:** Julie Augustin, Sándor Zsebők, László Zsolt Garamszegi.

**Funding acquisition:** Vilmos Bilicki, László Zsolt Garamszegi.

**Investigation:** Julie Augustin, Dorottya Kovács, Zoltán Soltész.

**Methodology:** Julie Augustin, Sándor Zsebők, László Zsolt Garamszegi.

**Project administration:** Vilmos Bilicki, László Zsolt Garamszegi.

**Resources:** Vilmos Bilicki, László Zsolt Garamszegi.

**Software:** Julie Augustin, Sándor Zsebők, Dorottya Kovács, Zoltán Jánki, András Bánhalmi, Zoltán Soltész, Péter Seffer.

**Supervision:** Sándor Zsebők, Vilmos Bilicki, László Zsolt Garamszegi.

**Validation:** Julie Augustin, Sándor Zsebők, László Zsolt Garamszegi.

**Visualization:** Julie Augustin.

**Writing – original draft:** Julie Augustin.

**Writing – review & editing:** Julie Augustin, Sándor Zsebők, László Zsolt Garamszegi.

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
