## [Decision Letter · Decision Letter 0]

4 Nov 2025

PONE-D-25-52971Proximate determinants of the frequency of mosquito sounds: separating species-specific effects from environmentally driven variations - implications for AI species recognitionPLOS ONE

Dear Dr. Augustin,

Thank you for submitting your manuscript to PLOS ONE. After careful consideration, we feel that it has merit but does not fully meet PLOS ONE’s publication criteria as it currently stands. Therefore, we invite you to submit a revised version of the manuscript that addresses the points raised during the review process.

We look forward to receiving your revised manuscript.

Kind regards,

**Muzafar Riyaz, Ph.D.**

Academic Editor

PLOS ONE

Journal Requirements:

4. Thank you for uploading your study's underlying data set. Unfortunately, the repository you have noted in your Data Availability statement does not qualify as an acceptable data repository according to PLOS's standards.

7. We note that Figure 1 in your submission contain map image which may be copyrighted. All PLOS content is published under the Creative Commons Attribution License (CC BY 4.0), which means that the manuscript, images, and Supporting Information files will be freely available online, and any third party is permitted to access, download, copy, distribute, and use these materials in any way, even commercially, with proper attribution. For these reasons, we cannot publish previously copyrighted maps or satellite images created using proprietary data, such as Google software (Google Maps, Street View, and Earth). For more information, see our copyright guidelines: http://journals.plos.org/plosone/s/licenses-and-copyright.

8. We are unable to open your Supporting Information file [File Name]. Please kindly revise as necessary and re-upload.

Reviewers' comments:

Reviewer's Responses to Questions

**Comments to the Author**

1. Is the manuscript technically sound, and do the data support the conclusions?

Reviewer #1: Yes

Reviewer #2: Yes

2. Has the statistical analysis been performed appropriately and rigorously? 

Reviewer #1: Yes

Reviewer #2: I Don't Know

3. Have the authors made all data underlying the findings in their manuscript fully available?

Reviewer #1: No

Reviewer #2: No

4. Is the manuscript presented in an intelligible fashion and written in standard English?

Reviewer #1: No

Reviewer #2: Yes

5. Review Comments to the Author

Reviewer #1: The manuscript “Environmental and biotic determinants of mosquito wingbeat frequency and implications for AI-based species recognition” is a timely and important study that addresses a central limitation in current mosquito acoustic monitoring: the lack of integration of environmental and biotic variability into species classification frameworks. The manuscript presents an impressive dataset (475 individuals from 15 species) and provides novel quantitative insights into within- and between-species repeatability of wingbeat frequency. The work is rigorous, relevant, and provides clear perspectives for future research, particularly regarding how biological and environmental covariates could improve AI-based classification of vector species.

The manuscript is scientifically strong and has clear potential for publication in PLOS ONE. However, revisions are necessary to improve clarity for a general audience, ensure methodological transparency, and align statements across sections. Once these issues are addressed, the paper will make an important contribution to bioacoustics and vector ecology by quantifying the influence of environmental and biotic variability on mosquito sounds and demonstrating its relevance for AI-based species recognition.

Strengths:

- Addresses a relevant and emerging intersection between bioacoustics and vector surveillance.

- Employs a well-chosen mixed-model framework with clear variance partitioning.

- Includes non-model mosquito species, expanding the taxonomic and ecological scope of acoustic studies.

- Provides actionable implications for improving AI-based mosquito recognition systems.

Specific Comments

1. Accessibility and scope of the Introduction

The Introduction is scientifically rich but currently too narrowly focused for the journal’s broad audience, i.e. briefly explain repeatability (statistical proportion of variance due to consistent among-individual or among-species differences) and why it matters for classification robustness and introduce vector species in simple terms (mosquitoes capable of transmitting pathogens).

Lines 41–70: Reorganize to introduce the ecological and applied relevance of mosquito acoustic monitoring before diving into technical details. For a general audience, it would help to first explain why acoustic species identification matters (e.g., vector surveillance & disease control).

Line 45: Briefly explain opto-acoustics for non-specialists

Lines 49–51: If they reach accuracy of 97% what is there to improve? And what is meant by underutilized and why is that important in this context? Clarify whether these results apply only under controlled laboratory conditions and why field deployment remains limited (e.g., lack of robustness, difficulty in implementation, or lack of trust). Strengthen the logical transition to “However…” in line 51.

Line 64: Quantify “data remain scarce”—for example, indicate the approximate number of studies or species covered (“data currently exist for ~6 studies on 2–3 model species”).

Tables 1 & 2: These tables are rich but create an impression that the topic is well explored, which contradicts the statement of scarcity. Consider moving them later in the Introduction or summarizing them graphically. A visual summary (e.g., species icons with circles representing investigated factors) would help illustrate the strong research bias toward Aedes aegypti.

2. Clarify the rationale for the focus on Aedes aegypti

The Introduction would benefit from a short statement explaining why Aedes aegypti dominates the literature (e.g., ease of laboratory maintenance, relevance as a disease vector). This contextualization will help readers understand the novelty of including less-studied species.

3. Methods

While the methods are thorough, several key steps need clarification to ensure reproducibility and transparency.

Line 127: Rephrase for clarity: “Wild-caught mosquitoes were recorded shortly after capture. Individuals reared from laboratory colonies did not differ in wingbeat frequency and were therefore included in the analysis.”

Lines 148–150: Clarify how acoustic stimulation works within a soundproof box. Does the operator reach into the box or use an external tool?

Table 3 (Age): Clarify that the age range reflects individuals bred from wild-collected eggs or larvae, not adult field captures. Currently it is unclear whether wild adults were recorded.

Lines 169–174: Move this paragraph earlier (before recordings) and expand on why estimating median age per cage is a reasonable approach.

Line 177: Specify the software environment of the “custom-made mosquito detection module” (e.g., R, Python, MATLAB). Indicate whether the code is publicly available and provide a repository link. If not, explain why.

Line 190: Explain selection criteria more explicitly—what was done when only one high-quality sound was available? Did usable sounds cluster temporally (e.g., early vs. late in recording)?

Line 192: Again, specify if analysis code is available.

Line 196: Explicitly name the extracted output parameter (e.g., “f1_freq = mean frequency of the first harmonic”).

Line 209: Explain what “different hierarchical levels” refers to (sounds within individuals, individuals within species) and why repeatability is the metric of interest. Define its possible range (0 = no consistency; 1 = complete consistency).

Line 211: Describe the specific parametric bootstrap procedure used (number of simulations, software function). If the scripts are not shared, reproducibility is limited—consider providing them as supplementary material.

Line 232: Adjust the Abstract to clarify that analyses were conducted on 10 species, though recordings were made for 15.

Table 4: Specify whether numbers represent all recorded individuals or those included in analyses.

Table 5: Add units (Hz), and consider presenting range and median in addition to mean ± SD. High SD values (often >10 % of the mean) merit brief comment.

Table 6: Add a short note in the Introduction explaining what repeatability values represent and how to interpret them (e.g., “values near 1 indicate high consistency across individuals or species”).

Line 269: Clarify why wing length was standardized within species and discuss potential implications of this choice on the non-significant result.

Lines 270–276 / Fig. 2: Provide a clearer description of random-slope results. Highlight that not all species respond similarly to temperature, and discuss how unequal sample sizes per species may influence slope estimates.

Line 284: Spell out “a good number of non-model species (n = 10 analyzed)” to emphasize novelty.

4. Results and Discussion

The Results are generally clear and statistically well presented, but some interpretation could be expanded for accessibility:

- Consider briefly summarizing expected vs. observed repeatability patterns (Table 6) for readers unfamiliar with the metric.

- The Discussion effectively highlights temperature and sex as key predictors but could benefit from a clearer paragraph summarizing the practical implications for AI-based classification (e.g., integrating temperature correction factors or including metadata in training).

- The final section on mate selection and individual quality is interesting but peripheral. Condense or explicitly link it to the study’s broader message about acoustic variability and consistency.

Reviewer #2: The authors present a study of the variation of mosquitoes' wingbeat frequency (WBF) accross species, sex, temperature and other environmental conditions. I found the study is intesresting because too many studies does not take into consideration the context where the mosquito WBF are recorded.

MAJOR COMMENTS

1) insufficient information on the "time of recording" parameter

Line 104-106, 181 : "due to the strong circadian variations in mosquitoes activity, we evaluated the impact of the time of recording"

 no methodological informaiton on this parameter, i.e. how time of recording was controlled and which ranges it took. What time were the mosquitoes recorded in the day and when was it as compared to the mosquito circadium time.

Did you record mosquitoes at a time independnat of wether they were day or night mosquitoes?

2) Insufficient information on acoustic processing

Line 148 / 181: not clear how the authors dealt with superimposed WBF when more than 1 mosquito was flying at a time

3) Insufficient information on acoustic recording

Line 148 / 181: not clear how the authors proceed to record the mosquitoes. More information is needed to know the behaviour of the mosquitoes during the recording, e.g.:

How was managed the opening of the soundproof box to make them fly and then record them? Did the authors wait for some time after the sonudproof box was closed?

How long time do the mosquito fly in the cage in general?

What are the mean duration of an long enough flight?

Was it full darkness in the soundproof chamber? If yes, what would be the effect on WBF / mosquito behaviour?

4) Another parameters that affects male WBF is the acoustic detection of the female (table 2, ...):

See the review book chapter you cite (49): male fundamental WBFs are observed to reach up to 1 000 Hz for short periods of time in swarms (Pantoja-Sanchez et al. (2019), Garcia Castillo et al., 2021; https://doi.org/10.1242/jeb.243535 , https://doi.org/10.1126/sciadv.abl4844). Then, the behaviour could also be a factor of WBF.

MINOR COMMENTS

5) line 78 "mating does not appear to alter wingbeat frequency". WBF is altered during matin.g I guess you meant "mated status"?

6) Line 140: "N=12" : not clear if it is the number of mosquitos used per cage.

7) Table 3: Usually, good temperature sensors have a precision of ~1°C, then having 1 digit precision in table 3 can be seen as an excess of precision: what is the tempreature accuracy of Voltcraft DL-210TH? Also it seems that Voltcraft DL-210TH is a datalogguer, not a sensor (not checked carefully). In this case, you should say which sensor was plugged to the data logguer and its temperature accurary.

Humidity sensor are even far less precise than temperature sensor and a 1 digit precision is not meaningfull.

8) Table 5, line 320: 2-digits for WBF are useless and give a wrong sense of precision which does not exist on . I would advice not to put any digits, and at least to remove the 2nd digit.

9) line 300: and 302: please gives the amplitude of change in Hz/°C and statistical test results; Discuss wether it could make any difference.

10) Given a species, are the slopes the same between males and females? This could be interesting in terms of mosquito hearing; indeed, males hear the difference frequency between they own WBF and that of the nearby female (distortion product from their antennae; see Warren et al 2009) ; this would tell us whether their hearing organ tuning woul need to thave the same temperature gradiant for instance. Indeed, hearing difference-tones cancels the effect of temperature when hearing each other, as suggested by Lapshin and

Vorontsov (2017).

11) line 388: the harmonic convergence theory is somewhow out of date (e.g. see the review chapter Feugere et al 2022; Somers et al 2022; Warren et al. (2009))

12) see also Villarreal et al., 2017 for effect of temperature

13) In the abstract, I would not highlight that female's WBF are lower than males' one, as it is really well established.

NB: I have not checked the data on figshare

6. PLOS authors have the option to publish the peer review history of their article (what does this mean? ). If published, this will include your full peer review and any attached files.

**Do you want your identity to be public for this peer review?** For information about this choice, including consent withdrawal, please see our Privacy Policy .

Reviewer #1: No

Reviewer #2: **Yes:**  Lionel Feugère

---

## [Author Response · Author response to Decision Letter 1]

16 Dec 2025

We thank the academic editor and reviewers for their interest in the manuscript and their comments. Please find below our responses to the comments. The lines indicated in this document correspond to the lines in the new version of the manuscript without track changes (“Manuscript”).

Academic Editor

Journal Requirements:

The manuscript’s format has been modified according to the guidelines (heading, figures, tables…).

The codes have been publicly shared either in a separate repository or in the figshare repository that also contains the data:

- Automatic detection of mosquito sounds: https://github.com/MosquiTUNE/Mosquito_sound_detection_methods/tree/main

- Sound measurements: https://figshare.com/s/084db2848ed6bad49a2e

- Statistical analysis: https://figshare.com/s/084db2848ed6bad49a2e

Because we only collected invertebrates, did not sample in private or protected areas or protected species, no permits were required. This was added in the methods, lines 176-179.

4. Thank you for uploading your study's underlying data set. Unfortunately, the repository you have noted in your Data Availability statement does not qualify as an acceptable data repository according to PLOS's standards.

The data was deposited on Figshare, and the data availability statement has been modified.

DOI: 10.6084/m9.figshare.30230581

Private link: https://figshare.com/s/084db2848ed6bad49a2e

The data availability statement has been updated with reference to the data repository on figshare.

The ethics statement was included in the methods section of the manuscript, lines 176-179

7. We note that Figure 1 in your submission contain map image which may be copyrighted. All PLOS content is published under the Creative Commons Attribution License (CC BY 4.0), which means that the manuscript, images, and Supporting Information files will be freely available online, and any third party is permitted to access, download, copy, distribute, and use these materials in any way, even commercially, with proper attribution. For these reasons, we cannot publish previously copyrighted maps or satellite images created using proprietary data, such as Google software (Google Maps, Street View, and Earth). For more information, see our copyright guidelines: http://journals.plos.org/plosone/s/licenses-and-copyright.

The map has been removed, and the collection data added as supplementary material (S3 Table). Figures and tables numbers and their references in the text have been adjusted accordingly.

8. We are unable to open your Supporting Information file [File Name]. Please kindly revise as necessary and re-upload.

The supporting information files have been revised and re-uploaded.

The specific published work that was mentioned by the reviewers was relevant and often already cited elsewhere in the manuscript.

Reviewer #1: The manuscript “Environmental and biotic determinants of mosquito wingbeat frequency and implications for AI-based species recognition” is a timely and important study that addresses a central limitation in current mosquito acoustic monitoring: the lack of integration of environmental and biotic variability into species classification frameworks. The manuscript presents an impressive dataset (475 individuals from 15 species) and provides novel quantitative insights into within- and between-species repeatability of wingbeat frequency. The work is rigorous, relevant, and provides clear perspectives for future research, particularly regarding how biological and environmental covariates could improve AI-based classification of vector species.

The manuscript is scientifically strong and has clear potential for publication in PLOS ONE. However, revisions are necessary to improve clarity for a general audience, ensure methodological transparency, and align statements across sections. Once these issues are addressed, the paper will make an important contribution to bioacoustics and vector ecology by quantifying the influence of environmental and biotic variability on mosquito sounds and demonstrating its relevance for AI-based species recognition.

Strengths:

- Addresses a relevant and emerging intersection between bioacoustics and vector surveillance.

- Employs a well-chosen mixed-model framework with clear variance partitioning.

- Includes non-model mosquito species, expanding the taxonomic and ecological scope of acoustic studies.

- Provides actionable implications for improving AI-based mosquito recognition systems.

Thank you for the appreciation.

Specific Comments

1. Accessibility and scope of the Introduction

The Introduction is scientifically rich but currently too narrowly focused for the journal’s broad audience, i.e. briefly explain repeatability (statistical proportion of variance due to consistent among-individual or among-species differences) and why it matters for classification robustness and introduce vector species in simple terms (mosquitoes capable of transmitting pathogens).

Repeatability and its importance for classification system was explained more in the last paragraph of the introduction, lines 138-144.

In addition, a sentence has been added in the beginning of the introduction to define mosquitoes as vector of pathogens, lines 43-45.

Lines 41–70: Reorganize to introduce the ecological and applied relevance of mosquito acoustic monitoring before diving into technical details. For a general audience, it would help to first explain why acoustic species identification matters (e.g., vector surveillance & disease control).

A sentence has been added to the beginning of the introduction to highlight the importance of mosquito monitoring, lines 45-47.

In addition, the importance of species identification has been added, lines 48-53.

Line 45: Briefly explain opto-acoustics for non-specialists

A brief explanation of the different type of data, including the opto-acoustics, has been added in the appropriate section, lines 58-62

Lines 49–51: If they reach accuracy of 97% what is there to improve? And what is meant by underutilized and why is that important in this context? Clarify whether these results apply only under controlled laboratory conditions and why field deployment remains limited (e.g., lack of robustness, difficulty in implementation, or lack of trust). Strengthen the logical transition to “However…” in line 51.

This section has been expanded to explain the conditions of high accuracy and why they are not yet deployed in the field, lines 66-75.

Line 64: Quantify “data remain scarce”—for example, indicate the approximate number of studies or species covered (“data currently exist for ~6 studies on 2–3 model species”).

The sentence has been modified as suggested, lines 91-92

Tables 1 & 2: These tables are rich but create an impression that the topic is well explored, which contradicts the statement of scarcity. Consider moving them later in the Introduction or summarizing them graphically. A visual summary (e.g., species icons with circles representing investigated factors) would help illustrate the strong research bias toward Aedes aegypti.

The text has been modified to highlight that the scarcity refers mostly to the assessment of impact of environmental factors, lines 89-91

Figures were created as suggested to replace the tables. The placement of Fig 1 and Fig 2 has been modified accordingly, to fit with the order they appear in the text. The titles and captions have also been modified. Subsequent tables and figures number in the text have been adjusted accordingly.

The original tables have been included as supplementary material (S1 Table and S2 Table) for readers that are interested in more of the specific effects.

2. Clarify the rationale for the focus on Aedes aegypti

The Introduction would benefit from a short statement explaining why Aedes aegypti dominates the literature (e.g., ease of laboratory maintenance, relevance as a disease vector). This contextualization will help readers understand the novelty of including less-studied species.

A short text has been added after the tables to explain the importance of the three most studied species, in particular Aedes aegypti, lines 103-106.

3. Methods

While the methods are thorough, several key steps need clarification to ensure reproducibility and transparency.

Line 127: Rephrase for clarity: “Wild-caught mosquitoes were recorded shortly after capture. Individuals reared from laboratory colonies did not differ in wingbeat frequency and were therefore included in the analysis.”

The text was modified as suggested, except for the first sentence that was left out because we did not record wild-caught adult mosquitoes, lines 162-164.

Lines 148–150: Clarify how acoustic stimulation works within a soundproof box. Does the operator reach into the box or use an external tool?

The protocol for the stimulation was clarified, lines 206-209.

Table 3 (Age): Clarify that the age range reflects individuals bred from wild-collected eggs or larvae, not adult field captures. Currently it is unclear whether wild adults were recorded.

The age reflects the number of days since adult emergence. It was clarified in the Table. In addition, the text was clarified in the methods to indicate that no field-caught adults were recorded, lines 162-164

Lines 169–174: Move this paragraph earlier (before recordings) and expand on why estimating median age per cage is a reasonable approach.

The paragraph was moved earlier (lines 135-158), and an explanation has been added on the relevance of using the median in this situation, lines 183-190.

Line 177: Specify the software environment of the “custom-made mosquito detection module” (e.g., R, Python, MATLAB). Indicate whether the code is publicly available and provide a repository link. If not, explain why.

The software environment was added to the description. In addition, the code was uploaded into a public repository, and its link has been added to the paragraph, lines 241-242

Line 190: Explain selection criteria more explicitly—what was done when only one high-quality sound was available? Did usable sounds cluster temporally (e.g., early vs. late in recording)?

This section has been expanded a little to explain in more details the selection process, lines 254-256.

Usable sounds did not cluster temporally and were distributed evenly throughout

---

## [Decision Letter · Decision Letter 1]

7 Jan 2026

PONE-D-25-52971R1Proximate determinants of the frequency of mosquito sounds: separating species-specific effects from environmentally driven variations - implications for AI species recognitionPLOS One

Dear Dr. Augustin,

Thank you for submitting your manuscript to PLOS ONE. After careful consideration, we feel that it has merit but does not fully meet PLOS ONE’s publication criteria as it currently stands. Therefore, we invite you to submit a revised version of the manuscript that addresses the points raised during the review process. Please submit your revised manuscript by Feb 21 2026 11:59PM. If you will need more time than this to complete your revisions, please reply to this message or contact the journal office at plosone@plos.org . Please include the following items when submitting your revised manuscript:

We look forward to receiving your revised manuscript.

Kind regards,

Muzafar Riyaz, Ph.D.

Academic Editor

PLOS One

Journal Requirements:

Reviewers' comments:

Reviewer's Responses to Questions

**Comments to the Author**

1. If the authors have adequately addressed your comments raised in a previous round of review and you feel that this manuscript is now acceptable for publication, you may indicate that here to bypass the “Comments to the Author” section, enter your conflict of interest statement in the “Confidential to Editor” section, and submit your "Accept" recommendation.

Reviewer #2: (No Response)

2. Is the manuscript technically sound, and do the data support the conclusions?

Reviewer #2: Yes

3. Has the statistical analysis been performed appropriately and rigorously? 

Reviewer #2: I Don't Know

4. Have the authors made all data underlying the findings in their manuscript fully available?

Reviewer #2: Yes

5. Is the manuscript presented in an intelligible fashion and written in standard English?

Reviewer #2: Yes

6. Review Comments to the Author

Reviewer #2: (No Response)

7. PLOS authors have the option to publish the peer review history of their article (what does this mean? ). If published, this will include your full peer review and any attached files.

**Do you want your identity to be public for this peer review?** For information about this choice, including consent withdrawal, please see our Privacy Policy .

Reviewer #2: No

---

## [Author Response · Author response to Decision Letter 2]

12 Jan 2026

The authors have make significant efforts to answer all my questions and I think the study worth being published, provided on adding the information given below on comment 3).

FOLLOWING-UP ON PREVIOUS COMMENTS

3) “The reduction of external sound was not as good as when the soundproof box was fully closed, but it was much better than no soundproof box. This was added in the methods, lines 206-209”

—> I cannot read this important information in lines 206-209

You are right, sorry about this oversight. A sentence addressing the box was added in the manuscript lines 210-211.

12) Is there a particular reason why Villarreal et al. 2017 is not cited in the litterature review for effect of temperature?

The Villarreal paper is cited in the S2 Table (and its data was used to build the corresponding Fig2). Since it was not the only paper that studied the impact of temperature, we chose to cite the table instead of the single reference. For example, line 108-110: “Overall, wingbeat frequency increases with temperature, and at high temperatures, wingbeat frequency also increases with humidity (S2 Table).”

NEW COMMENTS

14) “There was a strong variability in willingness to fly between individuals, with some mosquitoes flying non-stop throughout the 10min period, and some flying only for a second when stimulated. We included all mosquitoes that were observed flying in the analysis, regardless of their flight duration during the trial.”

—> If available, it could be interesting to have statistics on the duration of fly, and whether it is species specific, to check if circadium-dependant factors can affect this, e.g. whether night species only had short flights etc

We agree that investigating aspects of flight duration in more details would be an extremely interesting direction. However, in our design there is no robust way to assess flight duration in a standard way for each individual. Since our primary focus in this study was on characterizing mosquito vocalization, we applied a methodology that makes sound records. Based on such approach, we could extract info on flight duration for only those individuals that were flying close enough to the microphone. It is possible that the mosquito sound was not always detected by the microphone even when the mosquito was visibly flying, and this probability depended on the current position of the individual in the cage and the amplitude of its sound. Therefore, for the appropriate characterization of song duration we would have needed an approach that relies on video recordings as well in parallel to the sound recordings — which was initially not considered given the focus of the study. Therefore, we suggest not to present statistics on flight duration as assessed from the sound recordings, because this would represent a biased sample of individuals.

15) the cage is 10x10x10 cm3. Could this small dimension have an effect on the mosquito flight? Could it be a limitation of the study as compared to natural conditions?

While for small and medium species the cage appeared big enough to allow normal flying, it is true that for bigger species, the cage could affect their flight. However, this cage size allowed for the best trade-off between picking up the mosquito sound for small and quiet species, and allowing enough space for mosquitoes to fly naturally. This limitation was added into the manuscript, lines 192-196.

---

## [Decision Letter · Decision Letter 2]

1 Feb 2026

Proximate determinants of the frequency of mosquito sounds: separating species-specific effects from environmentally driven variations - implications for AI species recognition

PONE-D-25-52971R2

Dear Dr. Augustin,

We’re pleased to inform you that your manuscript has been judged scientifically suitable for publication and will be formally accepted for publication once it meets all outstanding technical requirements.

Kind regards,

Muzafar Riyaz, Ph.D.

Academic Editor

PLOS One

Additional Editor Comments (optional):

Reviewers' comments:

Reviewer's Responses to Questions

**Comments to the Author**

1. If the authors have adequately addressed your comments raised in a previous round of review and you feel that this manuscript is now acceptable for publication, you may indicate that here to bypass the “Comments to the Author” section, enter your conflict of interest statement in the “Confidential to Editor” section, and submit your "Accept" recommendation.

Reviewer #2: All comments have been addressed

2. Is the manuscript technically sound, and do the data support the conclusions?

Reviewer #2: Yes

3. Has the statistical analysis been performed appropriately and rigorously? 

Reviewer #2: I Don't Know

4. Have the authors made all data underlying the findings in their manuscript fully available?

Reviewer #2: Yes

5. Is the manuscript presented in an intelligible fashion and written in standard English?

Reviewer #2: Yes

6. Review Comments to the Author

Reviewer #2: (No Response)

7. PLOS authors have the option to publish the peer review history of their article (what does this mean? ). If published, this will include your full peer review and any attached files.

**Do you want your identity to be public for this peer review?** For information about this choice, including consent withdrawal, please see our Privacy Policy .

Reviewer #2: **Yes:**  Lionel Feugère

---

## [Editor Report · Acceptance letter]

PONE-D-25-52971R2

PLOS One

Dear Dr. Augustin,

I'm pleased to inform you that your manuscript has been deemed suitable for publication in PLOS One. Congratulations! Your manuscript is now being handed over to our production team.

Kind regards,

on behalf of

Dr. Muzafar Riyaz

Academic Editor

PLOS One